Analysis of differential expression of hair follicle tissue transcriptome in Hetian sheep undergoing different periodic changes

Chen Xueyan
Sun SunShuang
Sulaiman Yiming ysulaiman@xjau.edu.cn
College of Animal Science, Xinjiang Agricultural University , Urumqi , Xinjiang , China
Mata Fernando
Electronic publication date: 2024 Nov 25
Publication date: 2024
Volume: 12
Electronic Location ID: e18542
Received 2024 May 7; Accepted 2024 Oct 28
Copyright: ©2024 Chen et al.
Copyright year: 2024
Copyright holder: Chen et al.
License: This is an open access article distributed under the terms of the Creative Commons Attribution License, which permits unrestricted use, distribution, reproduction and adaptation in any medium and for any purpose provided that it is properly attributed. For attribution, the original author(s), title, publication source (PeerJ) and either DOI or URL of the article must be cited.
License URL: https://creativecommons.org/licenses/by/4.0/

Keywords: Hetian Sheep, lncRNAs, Hair follicles, mRNA

Funding: Xinjiang Natural Science Foundation Project “Preliminary analysis of Key microRNA regulatory network for carpet Wool Traits in Hetian Sheep” (2017D01A33) “Xinjiang Agriculture Research System” This work was supported by the Xinjiang Natural Science Foundation Project “Preliminary analysis of Key microRNA regulatory network for carpet Wool Traits in Hetian Sheep” (2017D01A33) and the “Xinjiang Agriculture Research System”. The funders had no role in study design, data collection and analysis, decision to publish, or preparation of the manuscript.

==============================
Background

This study provides new information on long non-coding RNA (lncRNA) and messenger RNA (mRNA) expression profiles in the hair follicles of Hetian sheep via the sequencing and analysis of the transcriptome of skin hair follicles during three periods of periodicity change. This is important for improving the quality of carpet wool, providing a preliminary basis for further research on the targeting relationship of these mRNAs and their target genes, and providing a scientific basis for marker-assisted selection of Hetian sheep.

Methods

The periodic variation of anagen (P I, May, n = 3), catagen (p II, October, n = 3), and telogen (p III, January, n = 3) of the skin hair follicle tissue of three Hetian sheep ewes were selected. Skin samples were collected from the right mid-side of each sampled sheep at three hair follicle developmental stages. The three sheep were used for each developmental stage as biological and technical replicates for transcriptome sequencing and analysis.

Results

The statistical power of this experimental design, calculated in RNASeqPower, was 0.92. Differential expression analysis revealed 81 lncRNAs that were differentially expressed (46 up-regulated and 35 down-regulated) and 129 mRNAs that were differentially expressed (46 up-regulated and 83 down-regulated) during the PI and PII periods. Between the PI and PIII periods, a total of 144 differentially expressed lncRNAs and 693 differentially expressed mRNAs were identified. Of these, 73 lncRNAs were significantly up-regulated and 71 were significantly down-regulated, while 474 mRNAs were significantly up-regulated and 219 were down-regulated. Additionally, a total of 87 lncRNAs were found to be differentially expressed, with 40 up-regulated and 47 down-regulated, along with 39 differentially expressed mRNAs (23 up-regulated and 16 down-regulated), between the PII and PIII stages. The functional assessment revealed that the mRNA expressed in the cells is related to the membrane, cell processes, metabolism, extracellular region, and other GO items. It is enriched in thyroid hormone synthesis, choline metabolism, cancer, AMPK, Hedgehog, and other signaling pathways.

Conclusion

A total of 2,286 lncRNAs (including 965 known and 1,321 novel lncRNAs) and 20,879 mRNAs were identified. These co-expressed differentially expressed genes could be used as candidate genes for studying the periodic changes of the hair follicles in Hetian sheep.

Introduction

Hetian sheep are a local breed of short-tailed heterogeneous semi-rough sheep that have economic advantages such as resistance to drought and heat, coarse feeding, and diseases (Shi et al., 2022; Yu et al., 2011). Their white and light-colored coats have long lines and high fiber elasticity that make them a high-quality raw material for weaving carpets and jacquard blankets (Wang et al., 2021; Yu et al., 2013). However, after many years of breeding and selection, the existing Hetian sheep population has poor wool qualities such as low wool yield, less fine bottom wool, and more dry and dead wool (Zhao et al., 2019a; Zhao et al., 2019b). Wool quality and yield are mainly determined by the hair follicle (Matsuzaki & Yoshizato, 1998), which has the characteristics of periodic growth. To improve wool yield and quality, it is essential to determine the periodicity of hair follicle changes in Hetian sheep and investigate the molecular mechanisms that regulate hair follicle development.

Hair follicles undergo periodic growth, which is mainly induced by changes in external temperature (Panteleyev, Paus & Christiano, 2000). The growth cycle is consistent with the annual season and includes three stages: anagen, catagen, and telogen (Yin, 2009). For mammals such as cattle and sheep, the growing season typically occurs between April and November of each year. The catagen phase usually takes place between December to January of the following year (Wu & Lu, 2015). Wang (2018) studied 4-month-old plain and mountainous Hetian sheep and used morphological and qPCR methods to determine that KAP4.2 was highly expressed in the skin. The total follicle density growth trend of Hetian sheep was found to be evident between February to August. In the growing season (February), the growth activity of hair follicles was weak, while the growth activity of hair follicles is better and strong in the 4 to 8 months group. The hair follicle density of the mountain-type Hetian sheep between June and August show significant growth because mountain-type Hetian sheep live at an altitude of 1,900 m, and temperatures vary widely. After years of observational studies, our research team determined that mountain-type Hetian sheep hair follicle growth occurs in May, the general catagen occurs in October, and January is the resting period (Wang, 2018).

It has been reported that a number of specific genes are selectively expressed or inhibited in specific cells of the hair follicle, and a large number of regulatory factors such as miRNA and long non-coding RNAs (lncRNAs) are involved in this process (Lei et al., 2012; Shanhe et al., 2017; Tian et al., 2024). Epidermal growth factor and nerve growth factor are involved in the process of hair follicle development. The WNT, Notch, and TGF-β signaling pathways may play an important role in the growth and development of hair follicles (Zhao et al., 2019a; Zhao et al., 2019b).

lncRNAs are a class of non-coding RNAs larger than 200 nt (Dunham, Kundaje & Aldred, 2012) and the most abundant non-coding RNA in eukaryotes. Depending on the cis-acting regulatory mode, lncRNAs may affect the transcription of its surrounding genes. In trans-acting mode, lncRNAs can regulate the expression of genes distally or in other chromosomes. This combination of genes affects the translation of downstream proteins, which play biological functions and participate in various biological processes such as cell differentiation, apoptosis, growth, and development (Mao et al., 2021; Marchese, Raimondi & Huarte, 2017). lncRNAs are important in regulating hair follicle development. Compared to telogen, 41 lncRNAs were upregulated and 157 lncRNAs were downregulated in anagen. Meanwhile, the effects of lncRNAs were detected with the target genes identified through cis- and trans-acting regulatory modes, suggesting that these lncRNAs have certain functions in the development and regulation of the hair follicle cycle. Fifteen significant differentially-expressed lncRNAs have been identified by analyzing the expression profile of the lncRNAs. These crucial differentially expressed lncRNAs may be involved in the initial molecular mechanism that regulates hair follicle development (Yue et al., 2016). Guo (2015) identified 6,127 lncRNAs in growing and resting cashmere using the RNA-Seq method, including 54 differentially expressed lncRNAs. These differentially expressed lncRNAs were found to regulate the periodic growth process of cashmere.

To date, no reports have been made on the screening and correlation analysis of differentially expressed lncRNAs during the periodic changes of Hetian sheep hair follicles. Additionally, the molecular mechanism behind Hetian wool growth and development, as well as the formation of high-quality carpet wool, remains unclear. This study utilized RNA-Seq technology to identify differentially expressed lncRNAs and mRNAs in the hair follicles of Hetian sheep during different periods of hair follicle periodic changes. A regulatory network was constructed, and potential biological functions of the differentially expressed lncRNAs were identified through bioinformatics analysis.

Materials & Methods

Ethics statement

The methods were performed in accordance with the guidelines of good experimental practices adopted by the Institute of Animal Husbandry. All experimental protocols were approved by the Animal Ethics Committee (AEC) of Xinjiang Agricultural University, China on April 20, 2020 (20200420).

Sample collection and RNA extraction

The research samples came from the Kunlun sheep farm in Yutian County, Xinjiang, China (′ − 39°29′ to 81°9′ to 82°51′E, 35°14′ to 39°29′N). Three female Hetian sheep aged 12 months under the same feeding and management conditions from the same flock were randomly selected. The feeding and management conditions were relatively the same for all the animals. Skin samples were collected from the right mid-side of each sampled sheep at the three hair follicle developmental stages (anagen, catagen, and telogen). The three sheep were used for each developmental stage as biological and technical replicates. After hair shearing and alcohol disinfection, approximately one cm2 of skin samples were collected from the same shoulder region of each sheep at different stages of hair follicle growth period, starting with anagen (P1, May, n = 3), then catagen (P2, October, n = 3), and telogen (P3, January of the following year, n = 3). Three pieces of back skin tissue from each sheep were taken without analgesia according to experimental protocols (20200420). The skin tissues were rinsed in ice-cold DEPC-treated water, cut into small pieces, submerged in RNAlater (ABI, Foster City, CA, USA), and frozen at −70 °C until RNA and total protein extraction. After each sampling, the ewes were well cared for in the later stage. The three sampled sheep were alive after sampling. All experimental procedures with sheep used in the present study were given prior approval by the Animal Ethics Committee (AEC) of Xinjiang Agricultural University under contract (20200420).

High-throughput sequencing and analysis of the transcriptome

A total of nine libraries in the three time periods mentioned above were constructed using Illumina Xten double terminal sequencing at Shanghai Jingzhou Gene Technology Co., Ltd. The quality of the raw sequencing data was evaluated, and the sequences were filtered using fastp (Chen et al., 2018). The raw data were uploaded to the NCBI database (https://www.ncbi.nlm.nih.gov/sra/PRJNA1095361). Hisat2 (Kim et al., 2019) was used to align the genome sequence, and the reference genome was Oar_V3.1.91 (https://useast.ensembl.org/Ovis_aries_texel/Info/Index; ftp://ftp.ensembl.org/pub/release-91/fasta/ovis_aries/dna/Ovis_aries.Oar_v3.1.dna.toplevel.fa.gz). StringTie (Pertea et al., 2016) was used to compare the fragments within each gene segment after alignment. The trimmed mean of M values (TMM) algorithm was then used for normalization, and finally, the FPKM value of each gene was calculated. Gffcompare (Pertea & Pertea, 2020) was used to compare the results obtained by StringTie splicing with the genome.

In the first application, the fragments within each gene segment after comparison were counted using StringTie, normalized using the TMM algorithm, and the FPKM value for each gene was calculated. The results were stored in an Excel file and the table gave the FPKM value for each gene or transcript in addition to the number of reads aligned to the gene (shown in the Supplementary Files). For sequences sequenced to specific genes in the genome, the software edgeR was used for statistical quantification, and the number of sequenced reads corresponding to each gene and homogenizing between samples was calculated, a process known as gene quantification. On this basis, the differential genes obtained from the comparison of the two groups were calculated based on the information of the experimental groups. This includes gene quantification and transcript quantification, and both coding transcripts (mRNA) and non-coding transcripts (lncRNA). For each sample, the expression of each gene is calculated using FPKM as the count quantity, which is a value that can be used for comparison between different samples. Count quantity represents the specific number of reads compared to each gene. It calculates the sequence alignment for each transcript using the transcript as a reference (Ioanna et al., 2018; Griffith-Jones et al., 2006).

Screening of differentially expressed lncRNAs and mRNA

The I, II, and III set samples represent anagen, regression, and telogen, respectively. PI, PII, and PIII represent the averages of three samples in the same period (01180635 I, 00617149 I, and 01180656 I, respectively) and were divided into three groups for comparison: PII vs. PI represented the control group, PII represented the treatment group, and PI was the reference group; PIII vs. PI represented the control group, PIII represented the treatment group, PI represented the reference group; and PIII vs. PII represented the control group, PIII represented the treatment group, and PII represented the reference group. Differential expression between sample treatment and reference groups was analyzed using EdgeR (Robinson, McCarthy & Smyth, 2010). The screening criteria were Q-value (corrected P-value) < 0.05, and the expression value was upregulated two-fold (fold-change ≥ 2) or downregulated 2-fold (fold-change ≤ 0.5).

Prediction of the differentially expressed lncRNA target genes

The interaction between the lncRNAs and mRNAs was identified by both cis- and trans-acting regulatory modes. The cis-acting target genes of lncRNAs were predicted by searching the mRNA upstream and downstream of lncRNAs within the range of 10 kb on the genome as the target gene of lncRNAs. The trans-target genes were predicted based on the principle of sequence complementation matching. The complementary mRNAs were compared with the lncRNAs using BLAST, and the thermodynamic parameter values of the lncRNA and mRNA complementary pairing were calculated using RNAplex software (Tafer & Hofacker, 2008). The results above the software threshold were selected as the target genes of the lncRNAs. The differentially expressed mRNA was isolated from the target genes of the differentially expressed lncRNAs. GO and KOBAS 2.0 (https://bio.tools/kobas) were used for annotation of differential genes (Chen et al., 2011). The part of differential mRNA in the target gene of differential lncRNAs was isolated, and the relationship of interaction was determined according to the target relationship of lncRNAs-mRNA. The Cystoscope software (Shannon et al., 2003) was used for visual analysis. The miRanda tools (http://mirtoolsgallery.tech/mirtoolsgallery/node/1055) were used for the co-expression analyses, and the first 400 interactions with a correlation coefficient greater than 0.7 and P < 0.05 were selected, while those with a correlation coefficient less than 400 were subjected to the actual conditions.

In order to avoid false positive results, the enrichment analysis result P was corrected by multiple tests (false discovery rate, FDR). When FDR ≤ 0.05, GO terms and pathways were defined as significant enrichment of candidate genes.

Quantitative real-time polymerase chain reaction

In order to verify the accuracy of transcriptome sequencing, 10 of the differentially expressed lncRNAs and 10 of the differentially expressed mRNAs were randomly selected, with large differential expression and high CPM values between PIII and PI periods detected for relative expression analysis using qRT-PCR. RT-qPCR was performed in 96-well plates (Thermo Fisher Scientific, Waltham, MA, USA) on the StepOnePlus system (Applied Biosystems, Waltham, MA, USA). Primer sequences and characteristics are shown in Table S3. The reaction mix was performed using 0.3 µl of FastStart Universal SYBR Green Master (Thermo Fisher Scientific), 1 µl of 10 µM primer mix, 1 µl of a diluted 1:10 cDNA, and water for a final volume of 20 µl. Cycling conditions were 95 °C for 1 minute, and 40 cycles of 95 °C for 5 s, and 60 °C for 40 s. All RT-qPCR experiments were performed using three biological and three technical replicates. The upstream primers for qRT-PCR were designed according to the mature sequences and were quantified in real time using the SYBR Green dye method and the expression of Actin as the internal reference gene. Three replicates were set up for each sample. The results were used to calculate the relative expression of target genes using the 2- ΔΔCt method, where ΔΔCt = Ct target gene—Ct internal reference gene, and ΔΔCt = ΔΔCt experimental group—ΔΔCt control group with 01180635 I as the control group. All the results were compared with the sequencing results, presented in Microsoft Excel 2016, and were uploaded as a separate zip file.

Results

Original data and quality control

After removing the raw sequencing data and passing through the filtering process, a total of 21,499,205 bp high quality data were obtained on average for each sample, the quality of the data after filtering the raw data was more than 86%, and the Q30 was more than 90% for all samples (Table 1 and Tables S1 and S2). The number of reads grows exponentially with respect to the coverage of the genome and then converges to a certain ratio of saturation.

Table 1 Clean statistics of raw reads.

Sample	Total reads	Clean reads	Clean ratio	no rRNA	rRNA ratio	no rRNA pair	
S00617149I	90,982,528	89,864,784	98.77%	89,759,340	0.12%	89,759,340	
S00617149II	96,468,902	95,843,912	99.35%	90,462,322	5.61%	90,462,322	
S00617149III	86,368,040	85,588,616	99.10%	84,823,360	0.89%	84,823,360	
S01180635I	82,175,600	81,447,962	99.11%	79,434,522	2.47%	79,434,522	
S01180635II	84,978,642	83,951,322	98.79%	83,198,370	0.90%	83,198,370	
S01180635III	86,625,880	85,687,400	98.92%	85,629,238	0.07%	85,629,238	
S01180656I	91,730,370	90,951,556	99.15%	87,969,762	3.28%	87,969,762	
S01180656II	86,157,106	85,329,020	99.04%	84,280,136	1.23%	84,280,136	
S01180656III	82,772,778	81,732,942	98.74%	80,996,134	0.90%	80,996,134	

Table 2 Cross-reference the information with all relevant databases.

Sample.ID	Clean.Reads	miRBase matureTotal.A.R	Rfam	GenomeTotal.A.R	Genome	Uniq. Reads	Uniq.A.R	Uniq.A.R.P	
			Total.A.R		Total.A.R.P				
S0617149I	24,734,835	9,190,105	19,088,781	21,241,263	85.88%	1,261,644	1,073,735	85.11%	
S0617149II	19,905,899	7,732,352	15,797,163	17,385,377	87.34%	1,047,837	924,456	88.23%	
S0617149III	20,835,097	8,425,210	17,151,450	18,176,324	87.24%	566,931	375,684	66.27%	
S1180635I	18,071,465	5,987,699	14,767,751	15,853,567	87.73%	574,681	464,704	80.86%	
S1180635II	23,247,853	9,133,227	19,268,227	20,326,162	87.43%	576,101	380,432	66.04%	
S1180635III	21,133,345	8,680,199	17,706,242	18,554,892	87.80%	487,585	324,435	66.54%	
S1180656I	19,437,266	5,773,534	12,501,434	14,892,836	76.62%	1,092,439	589,932	54.00%	
S1180656II	22,195,423	8,453,494	18,120,710	19,395,202	87.38%	790,493	608,143	76.93%	
S1180656III	23,481,915	10,189,875	20,041,164	21,070,478	89.73%	676,323	428,483	63.35%	

Sequence alignment

The statistical power of this experimental design, calculated in RNASeqPower, was 0.92. Clean reads were aligned to the miRBase (https://www.mirbase.org/), Rfam (http://rfam.org/), and Genome databases using Hisat2, and the alignment results are shown in Table 2. 8,173,966 reads were aligned to the miRBase database for each sample on average, and the average comparable reads for each sample were 37.90%. A total of 18,544,011 reads were aligned to the Genome database for each sample on average, and the average comparable reads for each sample were 86.35%, with 73.09% having only one unique alignment position on the reference sequence. A total of 17,160,325 reads were aligned to the Rfam database for each sample on average, and the average number of comparable reads for each sample was 79.8%. According to the alignment results, reads of different lengths were extracted for statistics, which could visually show the composition of the data under different lengths. Figure 1 illustrates the specific locations on the genome where the sequences obtained by sequencing were compared to the genome, i.e., their distribution in different functional regions of the genome.

Figure 1 Functional region distribution of the sequences on the genome.

By examining the expression correlation among the samples, we found that the square of the Pearson correlation coefficient (R2) was greater than 0.93. The results of gene quantification are shown in Figs. 2 and 3 and indicate that the expression patterns among the samples were very similar, with high test repeatability and low variation, which allowed for the next test analysis. Principal component analysis (PCA) is designed to assess a specific grouping of samples and determine whether the distribution of samples in the results is consistent with the experimental design groupings, and was used to demonstrate the relationship and variation among samples (Fig. 4).

Figure 2 The mRNA scatter diagram showing the differential expression of PII vs. PI.

Figure 3 Correlation test between samples.

Figure 4 Principal component analysis.

From the alignment results of the three databases, the small RNA fragments were classified and annotated, and their number was counted (uploaded as Supplemental Files). The proportion of comparison between all samples and the genome was found to be more than 98.83%, more than 98.30% of all samples had comparison pairs on the reference sequence, and the unique comparison rate on the reference sequence was 0.07–0.53%. As shown in Fig. 1, the sequences were mainly distributed in the alignment gene, coding, splice site, intron, non-coding, and intervening regions. The reads had the highest proportion in the gene region and the lowest proportion in the splicing site.

Screening of the differentially expressed lncRNAs and mRNAs

According to the screening threshold, a total of 81 differentially expressed lncRNAs were identified in the PI and PII stages (Fig. 5), including 46 upregulated lncRNAs and 35 downregulated lncRNAs. In PI and PIII, 144 differentially expressed lncRNAs were identified, including 73 significantly upregulated lncRNAs and 71 significantly downregulated lncRNAs. In PII and PIII, 87 differentially expressed lncRNAs were identified, including 40 upregulated and 47 downregulated lncRNAs.

Figure 5 LncRNAs volcano of P II vs PI.

A total of 129 differentially expressed mRNAs were identified in the PI and PII stages, including 46 upregulated and 83 downregulated mRNAs. In the PI and PIII stages, 693 differentially expressed mRNAs were identified, including 474 significantly upregulated mRNAs and 219 significantly downregulated mRNAs. Additionally, 39 differentially expressed mRNAs were identified at the PII and PIII stage, including 23 upregulated and 16 downregulated mRNAs. The KIF7 gene from the KIF family, KRT84, KRT15, and KRTAP11-1 from the KRT family, as well as HOXB5 from the HOX family, are associated with hair characteristics.

Prediction of the target genes of the lncRNAs

The prediction of cis targets for the identified 1,509 cis-acting lncRNAs resulted in 1,199 potential target genes. Additionally, trans targets were predicted for 627 trans-acting lncRNAs, resulting in 1,994 potential target genes. Previous studies have shown that many of the predicted target genes, such as KRT84, KRT25, and KRT15 of the KRT family; HOXA1, HOXB2, HOXD3 of the HOX family; and BMP3 of the BMP family are related to hair traits. Additionally, 958 target genes were predicted to differentially express lncRNAs at PI and PII, while 1,076 target genes were predicted to differentially express lncRNAs at PI and PIII. Furthermore, 795 target genes were predicted to be differentially expressed by the lncRNAs at PII and PIII.

Enrichment analysis of GO and KEGG

Our analysis predicted target genes and differentially expressed mRNAs, which were mainly annotated in the cell, membrane part, cellular process, metabolic process, extracellular region, and other GO items. Bar charts in Fig. 6 represent significantly enriched GO items in each period of comparison. The KEGG enrichment analysis revealed significant enrichment in thyroid hormone synthesis, choline metabolism in cancer, AMPK signaling pathway, and the Hedgehog signaling pathway. Figure 7 displays the KEGG function distribution of PIII vs. PI target gene and mRNA combined analyses.

Figure 6 The GO enrichment diagram of the combined analyses of the lncRNA target genes and mRNA with differential expression of PIII vs. PI.

Genes involved in biological processes are shown in red. Genes contained in the cellular component are shown in green. Genes included in the molecular function are shown in blue.

Figure 7 KEGG function distribution of the PIII vs. PI target gene and mRNA combined analysis.

The magnitude of the qvalue of the color response. Going from green to red indicates an increasingly significant degree of enrichment.

Differentially expressed lncRNAs and mRNA co-expression network

Figure 8 displays the co-expression network of the lncRNAs-mRNA between the comparison groups. Red dots represent differentially expressed lncrnas and blue dots represent differentially expressed mRNAs, with larger dots indicating more interactions they are involved in. This suggests that Hetian sheep miRNAs have multiple targets. Common genes were found in all three comparison periods, including differentially expressed ENSOARG00000007090 and MSTRG.25673.2 in lncRNAs, as well as differentially expressed CYP2F1 and MTERF4. These genes played a crucial role in altering the hair follicles of the sheep during the three periods and could be considered as potential candidate genes for further analysis. Additionally, genes with large dots and dense interaction lines in the network can also be analyzed as candidate genes.

Figure 8 The lncRNA-mRNA co-expression network between the PIII vs PI comparison groups.

Red dots represent differentially expressed lncRNAs and blue dots represent differentially expressed mRNAs, with larger dots indicating more interactions they are involved in.

qRT-PCR verification

The sequencing results were compared and graphically represented using Excel (uploaded as Supplemental Files) and the results are shown in Figs. 9 and 10. The expression trend was consistent with the RNA-Seq results, indicating the accuracy and reliability of the genome sequencing results.

Figure 9 Comparison of the qRT-PCR and RNA-Seq of lncRNAs.

Figure 10 Comparison of the mRNA qRT-PCR and RNA-seq.

Discussion

In recent years, there have been significant developments in high-throughput sequencing and genome research. The RNA-Seq technology has been widely used to discover and identify new lncRNAs in different species (Consortium SM-I, 2014). RNA-Seq technology has been used by researchers to identify lncRNAs in developing chicken muscle (Li et al., 2012). However, there have been few studies on lncRNA in Hetian sheep, a native breed of Xinjiang. To investigate the variation in transcriptome expression and skin tissue during the periodic changes of the hair follicle in Hetian sheep, we utilized high-throughput sequencing to analyze the differential expression of lncRNAs and mRNAs across three distinct periods. Additionally, we constructed a network regulation diagram. This research aims to provide a theoretical basis for the molecular mechanism of hair follicle development in Hetian sheep and offer a reference for molecular-assisted breeding.

Upon comparing the transcripts and differential expressions of lncRNAs and mRNAs, we discovered that the number of mRNAs exceeded that of lncRNAs. Additionally, lncRNAs were found to be tissue-specific and expressed at a lower level than the protein-coding genes (Mercer et al., 2008; Derrien et al., 2012; Kutter et al., 2012). The study identified 2,286 candidate lncRNA transcripts and 20,879 mRNA transcripts from three periods of periodic hair follicle changes. The expression levels of lncRNAs were lower than those of protein-coding genes. Additionally, the genes were found to express differently during different periods. The group comparing anagen and telogen showed the most differentially expressed genes, indicating tissue- and time-specific gene expression.

When comparing the expression of differentially expressed genes at each period, we found that there were more differentially expressed genes in the telogen and anagen phases than in the other two comparison groups. This suggests that more differentially expressed genes are involved in the transition from telogen to anagen. Compared to the anagen phase, the number of genes involved in telogen increased, and the number of upregulated genes was also consistent. Li et al. (2014) found that among the genes with different expression levels, most of the genes related to the cell cycle were upregulated in telogen. Guo (2015) discovered that among the 55 differentially expressed lncRNAs in the anagen and telogen of cashmere, 32 lncRNAs were upregulated and 23 lncRNAs were downregulated during telogen. This finding is consistent with previous research.

Research has confirmed that the Hedgehog signaling pathway’s sonic hedgehog can regulate the anagen phase of hair follicles and the transition from telogen to anagen (Yang et al., 2024). The transmission of signals from the cell surface to the MAPK signaling pathway in the nucleus is crucial for the periodic changes of the hair follicles (Drew et al., 2007). The co-expression analysis of differentially expressed lncRNAs and mRNA revealed significant enrichment in the AMPK and Hedgehog signaling pathways, which is consistent with previous studies. The studies suggest that lncRNAs may regulate downstream target mRNA expression through the pathway during periodic changes of hair follicles. Further research should verify the relationship between differentially expressed lncRNAs and mRNAs using the dual luciferase-reporter gene system and conduct genome verification on the mRNAs. Finally, this study selected the key candidate genes for Hetian wool traits to provide molecular-assisted markers for breeding Hetian sheep. The study provides a preliminary basis for more in-depth investigation of the relationship between lncRNAs and targeted mRNA in Hetian sheep. However, further analysis and verification are necessary to determine the specific regulatory mechanism.

Conclusion

This study identified 965 known lncRNAs and predicted 1,321 new lncRNAs through comparative analysis of transcriptional libraries obtained from sequencing Hetian sheep hair follicles at different stages of hair follicle growth period during anagen, regression, and telogen. Additionally, 20,879 mRNAs were identified, further enriching the sheep lncRNA and mRNA information database. The study found differential expression of the two at different periods. Additionally, bioinformatics tools were used to predict the network structure and function of key targets and pathways related to the Hetian wool traits of these sheep. This provides a new perspective for future studies.

Supplemental Information

Supplemental Information 1 The expression of lncRNA

the expression of each gene is calculated for each sample using FPKM as the count quantity, a value that can be used for comparisons between samples. count quantity represents the specific number of reads aligned to each gene. It calculates the sequence comparison for each transcript using the transcript as a reference. Gene Expression is annotated from the previous results file using the GO and KEGG databases, listing the relevant entries for each gene in both databases.

Supplemental Information 2 Expressions of mRNA and and Enrichment analysis

Supplemental Information 3 Expressions of genes

Supplemental Information 4 qRT-PCR Verification results

Supplemental Information 5 Amplification results of qRT-PCR Verification

Table S1 The quality control results for the sequencing data

Table S2 Mapping statistic

Table S3 The parameters of the primer sequences for qRT-PCR Verification

Supplemental Information 9 Author Checklist

Supplemental Information 10 MIQE checklist

We would like to thank all the participants of the study. We would also like to thank LEXIS for language editing.

Additional Information and Declarations

Competing Interests

Author Contributions

Animal Ethics

Data Availability

The authors declare there are no competing interests.

Xueyan Chen conceived and designed the experiments, performed the experiments, analyzed the data, prepared figures and/or tables, authored or reviewed drafts of the article, and approved the final draft.

SunShuang Sun conceived and designed the experiments, performed the experiments, analyzed the data, prepared figures and/or tables, authored or reviewed drafts of the article, and approved the final draft.

Yiming Sulaiman conceived and designed the experiments, performed the experiments, analyzed the data, prepared figures and/or tables, authored or reviewed drafts of the article, and approved the final draft.

The following information was supplied relating to ethical approvals (i.e., approving body and any reference numbers):

This study was authorized by the Animal Ethics Committee (AEC) of Xinjiang Agricultural University, China (20200420).

The following information was supplied regarding data availability:

The sequence reads are available at GenBank: PRJNA1095361.

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
