# Peer review of "Analysis of differential expression of hair follicle tissue transcriptome in Hetian sheep undergoing different periodic changes"

_PeerJ, doi:10.7717/peerj.18542_

## Round 0.1 · original submission · Major Revisions

· Academic Editor

Major Revisions

Please carefully read the feedback provided by the reviewers and submit the revised version of your paper.

Reviewer 1 ·

Basic reporting

Brief summary:
The manuscript submitted for peer review in PeerJ by Shuang,et al. entitled “Analysis of the differential expression of the hair follicle tissue transcriptome in Hetian sheep undergoing periodic changes at different periods”, aimed to study the different mRNas and LncRNAs pattern between different phases. The authors identified 2,286 lncRNAs and 20,879. The differential analysis allowed the authors to identified up-regulated and down regulated LncRNAs and mRNAs when comparing the transcriptome analysis between phases.


Although an English certificate was provided, the manuscript is written in a very simple, sometimes ambiguous English. It is necessary to review the work by improving the English which is of a very basic and scholastic style.

Literature references should be improved in specific sentences along the text, namely on those mentioned in general comments.

Figure 3 cannot be read. Most of the information given in tables is irrelevant to the proposed objectives of the study. Should be provided as suplementary material.

Authors must choose the most relevant information displayed in figures (E.g. 11-13). The remaining results should stay in text only.

Why did you choose to only display differentially expressed mRNAs and LncRNAs from I vs II comparison?

Did you compare the upregulated mRNAs and LncRNAs between the 3 comparison groups?

Experimental design

The research objectives are interesting, although the practicability is limited.

It is not entirely clear how did you randomly select the 10 LncRNAs and mRNAs to validate the sequencing.

Methods are described sufficiently. However, the text is confusing and should be rephrased.

Also, how did you collect the skin samples without analgesia?

Validity of the findings

Similar studies have already been published in different breeds/species. Thus, the experimental design isn´t new.

The conclusions are well stated and limited to the actual findings. Although the comparison between the three hair follicle phases decreases in importance throughout the article.

Additional comments

Line 64 – The sentence where you cite Wang is confusing and should be redone.
Line 67-73 – The whole idea is confusing and should be rephrased. Also, the authors mentioned observational studies. Is there any studies been published? Can you reference any of those statements regarding follicle phase in Hetian sheep.
Line 74-88 – Please cite references.
Line 78-80 – Please rephrase.
Line 82-83 – Please cite. It is not unanimous that lncRNAs are “generally” biological functional.
Line 108-122 – authors are already reporting results in the introduction. This section should be moved/corrected.
Line 144 – Why did you use an old sheep genome database?
Line 217 – confusing paragraph. Please rephrase. This section should be moved to Methods.
Line 231 – Authors should explain the results presented (statistical power 0.92) to the reader in discussion and not presenting them a website hyperlink…
Line 268-270 – Reference needed
Line 274 – is KEGG analysis based on human genome?
Figure 11-13 – legend in the figure doesn’t correspond to the description.
Line 339 – “and lncRNAs may regulate mRNA expression via these pathways to participate in periodic changes.” It is a presumption… They can also be by-products of these processes messaging.

·

Basic reporting

The manuscript is generally clear and professionally written, with a well-structured introduction that provides sufficient context and background. It effectively outlines the importance of Hetian sheep and the rationale for studying their hair follicles. The literature is well-referenced and relevant, covering key studies that provide a solid foundation for the research. The manuscript adheres to PeerJ standards and discipline norms. The figures included are relevant, of high quality, well-labeled, and appropriately described, which aids in the understanding of the results. Additionally, the raw data is supplied in accordance with PeerJ policy, with sequencing data properly deposited in the NCBI database. However, the results section could benefit from clearer presentation and additional details on gene validation, and minor language improvements are needed throughout the manuscript.

There are minor grammatical errors and instances of redundancy throughout the manuscript that should be addressed to enhance clarity. For example:

Line 30: Please change “The analysis of differential expression..” to “Differential expression analysis..”
Line 56: “….characteristic...” to “….characteristics…”
Line 119: Please revise the coordinates from eastern longitude 81°9 '-82°51', northern latitude 35°14 '-39°29' to 81°9' to 82°51' E, 35°14' to 39°29' N.
Line 196-197: Please revise the sentence. Unclear what the number “21499205” refers to.

Experimental design

The experimental design of this research is sound and well within the scope of PeerJ, presenting original primary research. The research question is well-defined, relevant, and meaningful, focusing on the expression profiles of lncRNAs and mRNAs in Hetian sheep hair follicles. This study fills an identified knowledge gap by providing new insights into the molecular mechanisms underlying hair follicle development in Hetian sheep. The investigation is rigorous and performed to a high technical and ethical standard, with the use of RNA-Seq for transcriptome analysis being well-justified and the experimental design robust. While the methods are described in sufficient detail, including additional information on the criteria for selecting differentially expressed genes and their validation would further enhance replicability.

Here are some minor comments:
Line 142: Provide details if raw reads were trimmed prior to alignment.
Line 144: Please provide the link to download the reference genome.

Validity of the findings

The underlying data provided are robust, and well-controlled, ensuring the reliability of the analysis. The conclusions are well-stated, effectively linked to the original research question, and firmly based on the supporting results. They thoughtfully discuss the implications of the findings, offering a clear understanding of the study's contributions to the field.

Figure 6 could be improved by modifying the x-axis to include genes that are highly up- or down-regulated. Currently, genes with log fold changes (LFC) greater than 10 are misrepresented at the extremities of the plot. Adjusting this would provide a more accurate representation of the data and improve the clarity of the figure.

For Figures 14 and 15, which compare qRT-PCR and RNA-Seq results for selected lncRNAs and mRNAs, the manuscript does not explicitly mention any statistical tests performed to validate these comparisons. While the figures show consistent expression trends, providing statistical measures, such as correlation coefficients and p-values, would enhance the credibility of the findings.

Additional comments

no comment

Reviewer 3 ·

Basic reporting

Image clarity could be improved. Tables need to be formatted to meet the requirements of the journal。

Experimental design

no comment

Validity of the findings

no comment

Additional comments

1.At the conclusion of the abstract, “These co-expressed differentially expressed genes could be used as candidate genes for studying the periodic changes of the hair follicles in Hetian Sheep”, this conclusion is unclear, and author should provide the gene ID. In addition, whether the co-expressed genes mentioned by the authors are genes that are co-expressed differentially expressed in all three periods or genes that are differentially expressed in one of the two periods.
2.The serial numbers of the tables are not in the right order, and it is also recommended that the QC results put the important ones in the body of the article and the rest of the results as attachment.
3.Line 163-171, the author do not make it clear how the differentally expressed analysis was done, such as whether p-value or corrected p-value was used, and what the differential expression multiplicity was. Line 173-176, the author used a correlation coefficient of more than 0.7 and P-value < 0.05 for co-expression analyses, and Why such thresholds are set.
4.The miRNAs appear in Figures 11-13, but the miRNA results do not appear in the article, so please explain.
5.Image clarity could be improved.
6.The second paragraph of the introduction needs to be corrected grammatically, and letter capitalisation needs to be checked and corrected.
7.Remove the excess "%" of L201.

Annotated reviews are not available for download in order to protect the identity of reviewers who chose to remain anonymous.

---

## Round 0.2 · Major Revisions

· Academic Editor

Major Revisions

Dear authors,

Please ensure you address all the issues raised by the reviewers before resubmission.

Reviewer 1 ·

Basic reporting

See below

Experimental design

See below

Validity of the findings

See below

Additional comments

I have read the corrected manuscript, and the authors made all the corrections according to my suggestions.

Also, the English language has improved a lot.

I have a few remaining comments:

• Legend of figure 11 seems to be wrong “The GO enrichment diagram of the combined analyses of the lncRNAs target genes and mRNA with deferential expression of PII vs. PII”

• Authors should review the sentence from line 59 “In the growing season (February) the hair follicle activity is weak (…)”

In the paragraph before, they state “The rest period typically falls between February and March of each year”

• Figure 1 is not that relevant since it is just a representation of the initial reads and consequent quality of the study.

The paper should be published after the authors follow up on the comments above.

·

Basic reporting

The figure numbers mentioned in the main text, the end of the manuscript, and the filenames of the actual figures are inconsistent. For example, Figure 6 is now labeled as Figure 9. However, Line 278 still states, "Figs. 5 and 6 are scatter and volcano diagrams, respectively, of the differentially expressed lncRNAs and mRNAs," which is inaccurate given the updated filenames.
Additionally, Figure 9 could be enhanced by modifying the x-axis to include genes that are highly up- or down-regulated. Currently, genes with log fold changes (LFC) greater than 10 are misrepresented at the extremities of the plot. Adjusting this would provide a more accurate representation of the data and enhance the clarity of the figure.

Overall, the main text needs to be carefully reviewed to ensure that the figure numbers correctly correspond to the figures referenced. There are multiple instances where the figure numbers in the text do not match the updated figure labels, which could lead to confusion and misinterpretation of the data.

Experimental design

no comment

Validity of the findings

no comment

Additional comments

• The results section is dense and could benefit from being more concise and easier to navigate.
• Include more information on the criteria for selecting differentially expressed genes and their validation processes.

Reviewer 3 ·

Basic reporting

no comment

Experimental design

no comment

Validity of the findings

no comment

Additional comments

no comment

---

## Round 0.3 · Major Revisions

· Academic Editor

Major Revisions

Dear authors,

Unfortunately, according to the reviewers comments, you were unable to address several of the listed issues in the previous round of peer review. Additionally, there was no response letter from the authors addressing the comments, which is a critical part of the revision process. As a result, I recommend that the authors submit a detailed response letter along with the revised manuscript to ensure that all previous concerns are adequately addressed. Unless this is provided, I believe the manuscript may need to be considered for rejection.

·

Basic reporting

Thank you for your revised submission. However, it appears that several key points raised in the previous round of comments have not been adequately addressed. These concerns are critical to the clarity and rigor of the manuscript and need to be resolved before further consideration. Please ensure that all comments from the earlier review are fully incorporated in your next revision.
Please provide the link to download the reference genome.
Figure 6/9 could be improved by modifying the x-axis to include genes that are highly up- or down-regulated. Currently, genes with log fold changes (LFC) greater than 10 are misrepresented at the extremities of the plot. Adjusting this would provide a more accurate representation of the data and improve the clarity of the figure.
The figure numbers mentioned in the main text, the end of the manuscript, and the filenames of the actual figures are inconsistent. For example, Figure 6 is now labeled as Figure 9. However, Line 278 still states, "Figs. 5 and 6 are scatter and volcano diagrams, respectively, of the differentially expressed lncRNAs and mRNAs," which is inaccurate given the updated filenames.
Additionally, Figure 9 could be enhanced by modifying the x-axis to include genes that are highly up- or down-regulated. Currently, genes with log fold changes (LFC) greater than 10 are misrepresented at the extremities of the plot. Adjusting this would provide a more accurate representation of the data and enhance the clarity of the figure.

Experimental design

NA

Validity of the findings

NA

---

## Round 0.4 · accepted · Accept

· Academic Editor

Accept

Dear authors,

Thanks for successfully addressing all the issues raised by the reviewers. Congratulations! your paper has now been accepted for publication on PeerJ.

·

Basic reporting

no comment

Experimental design

no comment

Validity of the findings

no comment

Additional comments

no comment